# Host–Viral Interactions in the Pathogenesis of Ulcerative Colitis

**DOI:** 10.3390/ijms221910851

**Published:** 2021-10-07

**Authors:** Torunn Bruland, Ann Elisabet Østvik, Arne Kristian Sandvik, Marianne Doré Hansen

**Affiliations:** 1Department of Clinical and Molecular Medicine, Norwegian University of Science and Technology, 7491 Trondheim, Norway; torunn.bruland@ntnu.no (T.B.); ann.e.ostvik@ntnu.no (A.E.Ø.); arne.sandvik@ntnu.no (A.K.S.); 2Department of Gastroenterology and Hepatology, Clinic of Medicine, St. Olav’s University Hospital, 7030 Trondheim, Norway; 3Centre of Molecular Inflammation Research, Norwegian University of Science and Technology, 7491 Trondheim, Norway; 4Department of Medical Microbiology, Clinic of Laboratory Medicine, St. Olav’s University Hospital, 7030 Trondheim, Norway

**Keywords:** ulcerative colitis, chronic inflammation, eukaryotic virus, innate immune responses

## Abstract

Ulcerative colitis is characterized by relapsing and remitting colonic mucosal inflammation. During the early stages of viral infection, innate immune defenses are activated, leading to the rapid release of cytokines and the subsequent initiation of downstream responses including inflammation. Previously, intestinal viruses were thought to be either detrimental or neutral to the host. However, persisting viruses may have a role as resident commensals and confer protective immunity during inflammation. On the other hand, the dysregulation of gut mucosal immune responses to viruses can trigger excessive, pathogenic inflammation. The purpose of this review is to discuss virus-induced innate immune responses that are at play in ulcerative colitis.

## 1. Introduction

Ulcerative colitis (UC), one main phenotypes of inflammatory bowel disease (IBD), is an idiopathic inflammatory disease of the human gastrointestinal tract that is characterized by relapsing and remitting inflammation limited to the colonic mucosa [1]. The exact etiology of UC is still not fully understood but is believed to involve dysregulated immune responses to environmental factors in genetically predisposed individuals leading to an inflamed colon [2]. This immune dysregulation was long attributed to an aberrant adaptive immunity, while more recent studies point to an essential role of the innate immune responses of the intestinal epithelium in the development and perpetuation of inflammation [3,4]. The colonic epithelium is a complex structure of several cell types, such as, e.g., absorptive cells, mucin-secreting goblet cells, hormone-secreting enteroendocrine cells, and minor cell types such as tuft cells, which primarily sense pathogens and release signaling substances (Figure 1). Collectively termed intestinal epithelial cells (IECs), the cells of this epithelial monolayer play an integral part in regulating mucosal antimicrobial defense and immunity [5,6].

The colonic monolayer of IECs together with tight cell junctions and a two-layered mucus cover acts both as a physical barrier, a sensor for pathogen- and damage-associated molecular patterns (PAMPs and DAMPs), and as an immune cell regulator [6,11]. Thus, the IEC sensing of PAMPs and DAMPs activates innate immune signaling pathways, resulting in the secretion of cytokines and chemokines, which secondarily regulate adaptive immunity. The dysregulation of these IEC-derived responses is considered to augment or prolong inflammatory responses during active UC [6] and is the focus of intensive research to unravel the pathogenetic mechanisms in UC.

Although viruses are among the most diverse and abundant biological entities, they have long been a neglected factor in studies on IBD pathogenesis. In addition to the well-established role of the enteric bacterial microbiota, recent studies have begun to reveal that enteric viruses are also critical for homeostatic regulation and disease progression, acting through both virus–bacteria (bacteriophages) and virus–host (eukaryotic virus) interactions [12,13,14,15,16]. The gastrointestinal tract is considered to be the organ that is the most exposed to viruses. In addition to bacteriophages, the enteric virome [15] consists of viruses with tropism for eukaryotic cells. Viruses are obligate intracellular “parasites” that rely on their host to replicate their genome and produce infectious progeny. For example, human norovirus has been shown to replicate in intestinal enteroendocrine cells [17]. Unlike bacteriophages, these eukaryotic viruses can affect host immune responses through direct interaction with innate immune sensors in the infected epithelial cells.

A diverse community consisting of both eukaryotic RNA and DNA viruses colonizes the gut mucosa [18,19]. It is estimated that the average human has ~8–12 chronic viral infections at any given time [20]. These viruses may cause symptomatic disease, or they may remain latent in healthy people. Emerging evidence indicates that these viruses play a role in human health and may, in turn, have beneficial and/or damaging effects on the host [15,19,21,22,23,24,25]. An unfavorable alteration of the gut virome composition has been implicated in chronic immune disorders, such as in the pathogenesis of IBD [15,26,27]. Several eukaryotic viruses have been explored as possible causative agents of IBD, including herpesviruses [28], rotaviruses [29], noroviruses [30], influenza viruses [31], and the measles virus [32]. Understanding virome–immune system interactions from a broad perspective may be particularly important in intestinal disease because the pathogen recognition receptors (PRRs) are probably simultaneously engaged by different viruses.

Multiple enteric virus infections have been shown to interfere with viral replication and the expression of cytokines and PRRs in intestinal epithelial cells [33]. Aberrant regulation of the immune response to infecting viruses may result in chronic inflammation. Chronic inflammation is also one of the hallmarks of cancer [34] and promotes all stages of tumorigenesis. Although the exact pathway is unknown, chronic inflammation in the colon is suggested to be a link between ulcerative colitis and colorectal cancer [35,36,37].

## 2. Distinct Eukaryotic Viruses in UC

Eukaryotic viruses, which start colonizing the gut mucosa early in life, belong to the *Adenoviridae, Anelloviridae, Astroviridae, Parvoviridae, Picornaviridae,* and *Picobirnaviridae* families, and their richness increases with age [18,19,38]. Some of these viruses are known to be pathogenic and cause disease in humans, e.g., norovirus (NoV), which causes gastroenteritis. Some studies have examined the relationship between NoV and flare-ups in UC patients [29,30,39,40]. Although results are contradictory, some of studies have found a positive association between NoV infection and flare-ups (reappearance of disease symptoms) [30,40]. Other viruses such human Anelloviruses (AV) are omnipresent and most probably infect the entire human population, either chronically or by continuous re-infection. No convincing examples have demonstrated viral clearance from infected individuals [41]. Their impact on human life is not yet known, but with no evidence of a disease association, a potential beneficial effect on human health is possible. As in the case of a balanced gut microbiome, which is needed for a healthy intestinal microenvironment, the AV population may be part of a personal virus flora that positively influences human physiology: “the beneficial virome” [41,42,43]. A high prevalence of AV is found in the feces of children with UC relative to the feces of controls [44], whereas others have found *Anelloviridae* to be more frequent in controls than in UC patients [45]. The family *Pneumoviridae* is found to be more abundant in UC patients than in controls [45]. As such, which are the main viral receptors and signal transduction pathways engaged by these eukaryotic enteric viruses in host intestinal epithelial cells, and what do we know about their contribution to UC pathogenesis? Below, we describe the current state of knowledge. We first give an overview of the possible virus receptors and signaling pathways involved and then discuss the aberrant regulation of innate immunity in UC and the cellular responses that are relevant for pathogenesis.

## 3. Viral Sensing Pattern Recognition Receptors and Activation of Innate Immunity in Intestinal Epithelium

The localization of PRRs on IECs, which are susceptible to viral infections, allows these cells to rapidly participate in the immune response to sustain immune tolerance and to prevent inflammation. Conversely, interference with their function in the IECs may contribute to the development and perturbation of inflammation in IBD [46]. The PRRs, which are mainly devoted to sensing viral infections, include members of the Toll-like receptor (TLR) family, the RIG-like receptors (RLR), and the nucleotide-binding and oligomerization domain (NOD)-like receptors (NLRs). The activation of all of these PRRs can trigger innate immunity through the induction of interferons and cytokines. (Figure 2).

TLR1–5 and 9 are expressed in IECs of both the small and large intestine while TLR6, 7, and 8 are only expressed in the human colon [48]. The typical viral sensing TLRs are TLR3, 7, 8, and 9. TLR3 recognizes double-stranded RNA (dsRNA) [49]. TLR7 and TLR8 detect single-stranded RNA (ssRNA) while TLR9 engages unmethylated CpG DNA [50]. TLR signaling is divided into two types of pathways: one of which is MyD88-dependent and the other of which MyD88-independent but TRIF-dependent. While TLRs are important for recognizing viral PAMPs in extracellular compartments and endosomes [51], the cytosolic retinoic acid-inducible gene I (RIG-I), similar to receptors, play a role in the recognition of viruses that replicate and reside within the cytosol. These comprise two main cytoplasmic sensor proteins, RIG-I and melanoma differentiation-associated gene 5 (MDA5), which are constitutively expressed in the IECs of the human colon [52]. RIG-I and MDA5 detect viral-mediated dsRNA structures or uncapped 5′ triphosphate ssRNA, which are generated during infection by both RNA and DNA viruses and activate a canonical signaling pathway via the signaling adaptor mitochondrial antiviral signaling (MAVS) [53,54,55,56,57,58,59].

NLRs are highly conserved cytosolic PRRs that perform critical functions in surveying the intracellular environment for the presence of infection, noxious substances, and metabolic perturbations. The sensing of these danger signals leads to their oligomerization into large macromolecular scaffolds (called inflammasomes) and the rapid deployment of effector signaling cascades to restore homeostasis. Upon activation, the oligomerized receptors form into multi-subunit wheel-shaped structures recruiting the inflammasome adaptor protein apoptosis-associated speck-like protein containing a caspase recruitment domain (ASC), which aggregate to form ASC specks and serve as activation points for caspase-1, which promotes the maturation of interleukin (IL)-1β and IL-18 [60].

## 4. Aberrant Regulation of Viral-Related Innate Immunity in Ulcerative Colitis

TLR-mediated innate immune dysfunction has recently been implicated in the pathogenesis of UC. In addition, TLRs and TLR-mediated signaling pathways play a role in the efficacy of treatment [61]. TLR3 is constitutively expressed in colonic epithelium [46,62,63]. In contrast to a study reporting the unchanged expression of TLR3 in active UC versus inactive UC [46], our group found that TLR3 protein expression was enhanced in active UC compared to in inactive UC and in healthy controls [62]. In addition, during active UC, TLR2, 4, 5, 8, and 9 gene expression is upregulated [64,65].

Results from animal studies report a role for RIG-I and MDA5 in colitis development although their specific involvement is still poorly understood [66]. Since RIG-I and MDA5 are cytosolic sensors of viral replication products, they have not been widely studied in the context of IBD. Even so, one study found RIG-I mRNA and protein levels to be dramatically reduced in the intestinal tissues of patients with Crohn’s disease but not in UC [67]. Loss-of-function variants in the *Interferon Induced With Helicase C Domain 1* (*IFIH1)* gene, which encodes MDA5, were identified in children with very early onset IBD (VEO-IBD), suggesting a role for impaired intestinal viral sensing in IBD pathogenesis [68].

Dysregulated inflammasome activation is linked to inflammatory disorders [69,70]. A diverse number of RNA and DNA viruses activate inflammasomes. The inflammasome NOD-like receptor family pyrin domain containing 6 (NLRP6) is expressed in mucosal tissues [71] and predominately in the small and large intestine, especially by absorptive enterocytes, colonic goblet cells, and myofibroblasts [72], and participates in the progression of intestinal inflammation and enteric pathogen infections, which is in addition to being pivotal for homeostatic mucin secretion from goblet cells. NLRP6 is shown to respond to the internal ligands leading to the release of anti-microbial peptides (AMPs) and mucus, further demonstrating a protective role [73,74,75]. Inflammatory signals such as tumor necrosis factor-α (TNF-α) or viral stimuli induce the transcription of NLRP6 [76]. NLRP6 is shown to bind viral RNA via the RNA helicase DEAH-Box Helicase 15 (DHX15) and to interact with the MAVS protein to induce type I/III interferons (IFNs) and IFN-stimulated genes (ISGs) upon the sensing of positive-sense single-stranded RNA viruses [76], causing IL-18 secretion in IECs [77,78]. DHX15 plays a critical role in sensing enteric RNA viruses in IECs and in controlling intestinal inflammation [77]. Protein levels of DHX15 are reported to be reduced in UC patients [78], which can render the intestinal epithelial cells more susceptible to the inflammation caused by enteric RNA viruses due to the reduced production of IFN-β, IFN-λ3, and IL-18 [77]. No significant changes in the gene expression of NLRP6 have been demonstrated in UC patients [79,80]. The expression of the negative regulator of NLRP6 inflammasome, cylindromatosis (CYLD), has been shown to prevent excessive IL-18 levels. In the colonic mucosa of UC patients, the expression of CYLD is downregulated and is negatively correlated with IL-18 expression [79]. This suggests that the regulatory mechanisms inhibiting the excessive activation of NLRP6-mediated inflammation are defective in UC patients [73].

The NLRP3 inflammasome is one of the best-characterized inflammasomes and is activated by a diverse number of stimuli, including both RNA and DNA viruses and K^+^ efflux induced by lytic cell death triggered by viral replication [81]. It has been shown not only to be a crucial mediator for host defense but also as a critical regulator of intestinal homeostasis [82,83]. During inflammation of the colon, the NLRP3 inflammasome manages innate immune responses, contributing to the ongoing inflammation and the disruption of the mucosal barrier through the modification of tight junction proteins and cell apoptosis [84,85]. During the active UC mucosal mRNA expression of NLRP3, the inflammasome components NLRP3, IL-1β, ASC, and Caspase-1 are increased, correlating with disease activity [86].

NLRP1 is expressed by a variety of cell types, including the epithelial structures of the colon [87]. Human NLRP1 is a direct sensor of dsRNA and thus RNA virus infection [88]. Microarray analysis of inflamed human colon biopsies from patients with UC showed a significant increase in NLRP1 gene expression compared to healthy controls, which could be associated with increased IFN-γ [89]. NLRP9b, which is specifically expressed in IECs, restricts the intracellular replication of the rotavirus in IECs by binding the dsRNA viral genome via RNA helicase DHX9, leading to inflammasome formation, the release of active IL-18, and the pyroptosis of infected cells [90]. NLRP6 and NLRP9b, with their different regional expression, are suggested to cooperate in the defense against enteric viruses with distinct tropism [16]. NLRP7 has been linked to innate immune signaling, but its precise role is still controversial, as it has been shown to both positively and negatively affect inflammasome responses [91]. Recently, a significant association between the low-frequency NLRP7 S361L variant and an increased risk of developing UC was identified [92]. However, any functional insight on how this variant contributes to immune system dysfunction is still elusive.

## 5. Effector Factors Downstream of Virus-Activated PRRs Relevant for Ulcerative Colitis Pathogenesis

The binding of PAMPs to the above-mentioned PRRs initiates a number of intracellular signaling cascades via, e.g., the transcription factor complexes of the nuclear factor kappa-light-chain-enhancer of activated B cells (NF-kB) or interferon regulatory factors (IRFs). The activation of NF-kB in IECs induces the secretion of interleukins and cytokines, whereas IRFs are more involved in the production of IFNs and the induction of ISGs. Cytokines have a crucial role in the pathogenesis of UC, where they control multiple aspects of the inflammatory response. Cytokines can positively or negatively affect the intestinal epithelial barrier integrity by driving or inhibiting critical epithelial cell functions such as proliferation, apoptosis, and appropriate epithelial barrier permeability. In particular, the imbalance between pro-inflammatory and anti-inflammatory cytokines that occurs in UC impedes the resolution of inflammation and instead leads to disease perpetuation and tissue destruction. These cytokines can be derived from resident innate or adaptive immune cells, infiltrating inflammatory cells, or intestinal epithelial cells themselves. The role of these cytokines has been reviewed elsewhere [93]. Here, we will discuss some of these cytokines that are related to the innate immune system that may be induced by viruses.

Epithelial cells are a major port of entry for many viruses, but the molecular networks that protect barrier surfaces against viral infections are incompletely understood. IFNs are a class of cytokines that are produced and secreted upon infection, by viruses in particular. Viral infections induce a simultaneous production of type I (IFN-α and IFN-β) and type III (IFN-λ) interferons. All nucleated cells are believed to respond to IFN type I, while IFN type III responses are largely confined to the epithelium due to the specific expression of the IFN-λ receptor on epithelial cells [94]. As such, IFN-λ has recently emerged as a key player in mucosal immunity, especially in the gastrointestinal tract [95]. The type III IFN pathway may tune the gut immune response better than type I IFN but can be negatively controlled during IBD through the dysregulation of viral-sensing PRRs [67,96,97]. It has been well established that epithelial cells are especially responsive to type III IFN, which strengthens the mucosal barrier and prevents viral entry and infection. Data illuminating the role of type III IFN signaling in the gut are generally scarce, except for in some viral infection and colitis models, where type III IFN signaling is mostly protective [98]. Although IFNs signal through different IFN receptors, their downstream transcriptional responses are similar, as they both induce a large number of ISGs [99].

In a recent study from our group, we reported the increased expression of several ISGs in IECs from patients with active IBD [96], including ISG15 —a ubiquitin-like protein that is known to be highly upregulated during the initial stages of viral infections [100,101]. We also showed that human colon-derived IECs (i.e., 3D colonoids) release free immunomodulatory ISG15 upon extracellular stimulation with the TLR3 ligand poly (I:C). In line with studies from other researchers [101,102], we showed that ISG15 enhanced the release of IBD-relevant cytokines such as CXCL1, CXCL5, CXCL8, CCL20, IL1, IL6, TNF, and IFNγ from immune cells. These results and previous studies from our group showing the TLR3-mediated release of C-X-C motif chemokine ligand 10 (CXCL10) [103] and C-C motif ligand 20 (CCL20) [104] from colonic IECs indicate that, e.g., a dsRNA virus can release proteins from IECs, recruiting immune cells to prolong inflammatory responses during active IBD. Our findings that type I IFN signature genes in IECs are upregulated in IECs during active IBD also suggest that responses to viruses might be involved in the pathogenesis.

Others have found an increase in IL28A expression (one of the type III IFNs) in the colonic epithelium of patients with UC where it induces the phosphorylation of STAT1, leading to epithelial cell proliferation in patient-derived organoids [105]. In immune-deficient mice, viral complementation caused gut-specific protection against two major enteric pathogens, norovirus and rotavirus, via the specific elevation of the type III IFN immune response in gut epithelial cells. Of note, neither type I nor II IFNs were elevated, indicating a specific and compartmentalized IFN type III response at the mucosal barrier. Thus, an element of the enteric virome may confer innate immune-mediated protection without triggering adverse systemic inflammation [99,106].

IL-18 is a member of the IL-1 family of cytokines and is synthesized as an inactive precursor requiring processing by caspase-1 into an active cytokine by an inflammasome. The IL-18 precursor is present in the epithelial cells of the entire gastrointestinal tract [107]. The activity of IL-18 is balanced by the presence of a high affinity, naturally occurring IL-18 binding protein (IL-18BP). Increased disease severity can be associated with an imbalance of IL-18 to IL-18BP, elevating the levels of free IL-18 in the circulation [107]. Elevated levels of IL-18 are found in patients with UC [108]. On the other hand, IL-18 is an early trigger for tissue repair [109], and its activation by the inflammasomes in the IEC layer should aid in maintaining homeostasis unless high disease activity, which disrupts the epithelial barrier, inducing inflammasome activation, which may increase mucosal inflammation [110]. Thus, IL-18 has both a protective and detrimental role in colonic inflammation. A basal level of IL-18 in the colonic mucosa is required to maintain barrier integrity since the complete loss of IL-18 predisposes mice to intestinal epithelial damage, fostering an altered inflammatory environment that potentiates intestinal tumor formation [111,112]. On the other hand, elevated IL-18 levels promote inflammation and intestinal damage through goblet cell dysfunction, causing a breakdown of the mucosal barrier [108]. Increased NLRP6 inflammasome activity and IL-18 levels in the colonic mucosa due to deficiencies in the regulatory mechanisms controlled by CYLD, result in severe colitis in mice [79]. IL-18 may confer protection against colitis-associated inflammation [113,114] and neoplasia [109,111] by modulating the permeability of the intestinal epithelium [113,114], the production of antimicrobial peptides [85], and the activation levels of the tumor suppressors IFN-γ and STAT1 [109].

Interleukin-33 (IL-33) [115] is a nuclear cytokine that acts as an “alarmin” in response to the cellular damage induced by stress or by infection [116,117]. IL-33 is constitutively expressed in epithelial tissue, particularly in tissue barrier sites where this cytokine contributes to the maintenance of mechanical barriers [115,116,118,119,120]. It does not require processing by an inflammasome for biological activity but rather is inactivated by caspase cleavage [121]. Following translation, it is stored as a full-length, biologically active molecule in the nucleus, where it binds chromatin [122]. By its sequestration in the nucleus, IL-33 can act as a transcriptional regulator by binding the p65 subunit of NF-kB to activate endothelial cells [123]. Following cell lysis through destructive mechanisms, IL-33 acts as an early notifier of damage through the recruitment of neutrophils, eosinophils, natural killer cells, and by amplifying a type 2 response in order to initiate fibrosis and wound healing [124,125]. Increased levels of IL-33 have been reported in biopsies from patients with active UC compared to healthy controls [126]. IL-33 expression can be induced through TLR3, possibly as a protective mechanism of the host [127].

## 6. Viral-Induced Goblet Cell Dysfunction

UC has been associated with a dysfunctional colonic mucus layer and a reduced number of mucin-producing goblet cells [7,128] that can cause infiltration of the microbiota due to increased contact with the epithelium, promoting further inflammation [129]. Analysis of biopsies from UC patients revealed a thinner mucus layer, especially in inflamed areas [130]. Goblet cell function has also been proposed to be impaired in UC [131,132,133]. Several studies report reduced levels of MUC2 in active UC compared to UC in remission and to healthy controls [134,135,136,137,138]. This reduction was either related to the ineffective translation of MUC2 mRNA in active UC [134,136], reduced differentiation of goblet cells [133,135,137], or poorly sulfated MUC2 protein, which is more susceptible to degradation [138,139]. The impaired induction of the goblet cell differentiation factors Hath1 and KLF4 during inflammation could explain relative goblet cell depletion and deficient mucin induction in active UC [133].

Goblet cells can respond rapidly to infection but must balance this response with maintaining homeostasis. Goblet cell defenses against bacteria and parasites have been characterized, while responses to viral infection are less known. Enterovirus 71 (EV71) infects intestinal epithelial cells, and specifically goblet cells, where it reduces the expression of goblet cell-derived mucins MUC1 and MUC2, suggesting a viral-induced alteration of goblet cell function [140]. In addition, the human adenovirus has also been shown to preferentially infect goblet cells, although this preference was strain-specific. Adenovirus species C showed goblet cell tropism, while species F did not [141]. Furthermore, astrovirus VA1 (AstV-VA1) also infects goblet cells as well as other epithelial cell types [142]. Moreover, subpopulations of goblet cells in the gut mucosa have been identified [143]. Reduced numbers of a specific subtype called intercrypt goblet cells (icGC) have been linked to a loss of intercrypt mucus and a defective mucus structure in UC patients, both in patients in remission and in patients with active disease, resulting in exposed areas of surface intestinal epithelium [143]. Sentinel goblet cells, which are situated at the entrance to the colonic crypt, may sense non-specific TLR ligands and may trigger intracellular NLRP6 activation, resulting in mucin exocytosis to flush out pathogens [8,75].

Evidence from animal studies has shown that murine astrovirus (MuAstV) infects actively secreting goblet cells and might benefit from causing an increased mucus secretion in response to the infection of goblet cells by aided egress or dissemination [144]. Such mucus secretion may lead to goblet cell exhaustion, with a weakened mucus barrier leading to inflammation or secondary bacterial infection [145]. Rotavirus infection in mice causes the apoptotic death of enterocytes, leading to decreased numbers of goblet cells as a result of delayed intestinal repair [146]. Transmissible gastroenteritis virus infects Paneth cells in pigs, causing a loss of Notch signals, which induces an increase in goblet cell numbers and mucus production [147]. Increased mucus production is beneficial for the virus due to binding to the sialic acid-rich MUC facilitating receptor interaction [148,149]. Taken together, this suggests that enteric viruses are able to cause substantial changes in goblet cell number, differentiation, and function.

## 7. Clinical Implications of Host–Virus Interaction: Challenges and Future Perspectives

Treatments that are currently available for the management of IBD include immune-modulating drugs. With these therapies, infective complications have emerged as a safety concern. Primary infection with the Herpesvirus family members Cytomegalovirus (CMV), Epstein–Barr virus, and Herpes virus 6 may induce colonic inflammation in both healthy and immunocompromised individuals [150,151,152,153]. However, the clinical significance of viruses other than CMV in patients with IBD is unknown. CMV reactivation may take place in patients with UC receiving potent immunosuppression treatments such as, e.g., high-dose/long-term corticosteroids and anti-TNFs, and implies a worse long-term prognosis, even if successfully treated with antiviral agents [154]. Since the reactivation of cytomegalovirus infection is seen in acute severe UC, the European Crohn’s and Colitis Organisation (ECCO) guidelines recommend antiviral treatment in such cases [23]. Recent studies have shown that SARS-CoV-2 can enter the host by binding to the enterocyte/colonocyte-expressed Angiotensin-Converting Enzyme 2 (ACE2) receptor. ACE2 receptors and inflammatory pathways are often found upregulated in active UC [155]. However, the clinical consensus is that UC does not represent an independent risk factor for contracting SARS-CoV-2 nor for a more severe COVID-19 disease course [156]. Reports of de novo UC after SARS-CoV-2 infection are few and only anecdotal [157], and it is presently not known whether the virus itself may induce a flare-up in patients with inactive UC.

As pointed out earlier, many viral families start colonizing the gut mucosa early in life, and their richness increases with age. Some of these viruses are known to be pathogenic and cause disease in humans, while others are omnipresent and infect a large part of the human population without any evidence of disease association. The characterization of what constitutes a healthy or diseased virome is still in its infancy [158]. The complete eukaryotic virome colonizing the intestinal mucosa of UC patients still needs to be defined. The lack of studies regarding the human virome is due to our inability to readily culture or detect them. For most viruses, the results of the initial infection can vary widely depending on the site of entry, the cell types that are infected, and the innate immune responses of the infected cell. The specificity of a given virus for a cell type (cellular tropism) is an important factor in determining the outcome of viral infection. Which cell type different viral groups infect to actively replicate, disseminate, or rather establish latent infection in may be important for their pathogenicity. We do not yet know the eukaryotic host cell of most viruses, and there is no universal 16S ribosomal RNA equivalent, in which is the case bacteria and that allowing for rapid taxonomic characterization. Technologies such as metagenomics have only recently enabled the identification of viruses in healthy human tissues [159]. Although several eukaryotic viruses have been explored as possible causative agents of IBD (paragraph 2), we are still far from being able to identify the specific viruses that are involved in UC pathogenesis. On the other hand, a specific composition of resident viruses in the intestinal mucosa may be essential for immune maturation and may be required for healthy gut immunity. Details on their precise roles in either beneficial or harmful immunomodulation are needed in addition to the identification of triggers for immune responses and the mechanisms involving epithelial homeostasis and wound repair. Increased understanding of the role played by viruses in IBD can prove to be fundamental for developing interventional and therapeutic strategies, helping us move towards precision medicine.

Overall, several recent findings have indicated that responses to certain viruses might be involved in UC pathogenesis. As described previously, other researchers as well as ourselves have shown the upregulation of central effector factors downstream of potential virus-activated PRRs during active IBD, suggesting the involvement of virus. Of special interest is the ability of enteric viruses to cause substantial changes in goblet cell number, differentiation, and function. In addition, the dysregulation of PRRs, either at the mRNA or protein level, or the inhibition of regulatory mechanisms in IBD patients may cause impaired intestinal viral sensing, contributing to pathogenesis. Our group recently reported the increased expression of several ISGs in IECs from patients with active IBD, which, together with the enhanced release of IBD-relevant cytokines, indicates that, e.g., a dsRNA virus can induce the release of proteins from IECs to recruit immune cells and prolong inflammatory responses during active IBD. Interferons are a class of cytokines that are produced and secreted, particularly upon viral infections. Our finding that the IFN signature genes are upregulated in IECs during active IBD suggests that responses to viruses might be involved in the pathogenesis of IBD. IFN type III has been shown to be primarily important for antiviral defenses at mucosal barrier sites, but there are conflicting reports as to whether this IFN type induces proliferation or epithelial barrier dysfunction in UC patients.

The ultimate goal is to translate findings from virome research into diagnostic and therapeutic opportunities. Furthermore, enteric infections (e.g., virus infection) and non-infectious flare (reappearance of disease symptoms) may elicit similar clinical, endoscopic, and histological findings, suggesting a future need for more refined diagnostic viral detection than what is clinically routine today. The complexity and variability of IBD disease pathogenesis will likely require a personalized and multidimensional treatment approach. The identification of factors contributing to excessive inflammation in the gut epithelium has the potential to improve treatment options for some of these patient groups.

## 8. Concluding Remarks

Even though viruses are among the most diverse and abundant biological entities, they have long been a neglected factor in studies on IBD pathogenesis. A growing amount of evidence demonstrates that the gut virome is not just detrimental or a neutral bystander in the gastrointestinal tract but instead actively participates in establishing homeostasis and disease development.

Our understanding of the interplay between the human virome and the immune system has improved greatly in recent years. However, most studies rely on single virus infection models. Expansion into coinfection studies or a greater diversity of virus types that also include commensal viruses will generate a better understanding of the complex interplay when several viral sensing PRRs are engaged simultaneously. Understanding PRR signaling and innate immunity in UC pathogenesis are becoming increasingly imperative to identify components of the signaling pathways as treatment targets.

## Figures and Tables

**Figure 1 ijms-22-10851-f001:**
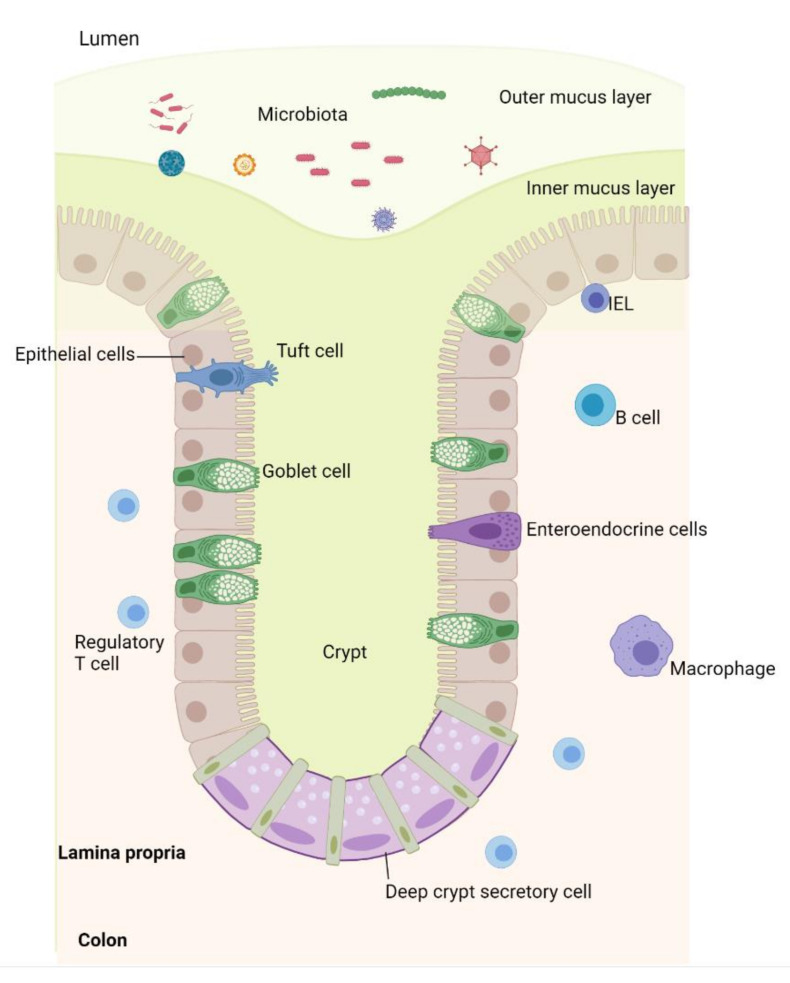
Colon epithelium. The deeper colonic mucosa is separated from the environment by a single layer of intestinal epithelial cells (IECs) constituting a physical and functional barrier. The IECs originate from the crypt base stem cells, which divide to give rise to more proliferative daughter cells. Most daughter IECs move upwards along the crypt and differentiate owing to a decreasing gradient of growth factors. IECs that reach the tip of the crypt undergo apoptosis and are then shed to the lumen. This entire cycle typically lasts 4–5 days. Throughout this migration, IECs differentiate into absorptive enterocytes, mucus-producing goblet cells, hormone-secreting enteroendocrine cells, tuft cells, or deep crypt secretory cells. A firm inner mucus layer reduces exposure to microorganisms. Intraepithelial lymphocytes (IELs), B cells, IgA-producing plasma cells, macrophages, and T cells reside in the lamina propria, contributing to and maintaining a hyporesponsive state. The microbiota may engage Pattern Recognition Receptors (PRRs) expressed on the IECs and may induce innate immune responses [6,7,8]. Mucins are the primary organic components of mucus, the properties of which are governed by the mucin structure. MUC2 is the most prominently expressed mucin in the colon [9,10]. Illustration created by MDH using Biorender.com. https://biorender.com/ (accessed on 30 August 2021).

**Figure 2 ijms-22-10851-f002:**
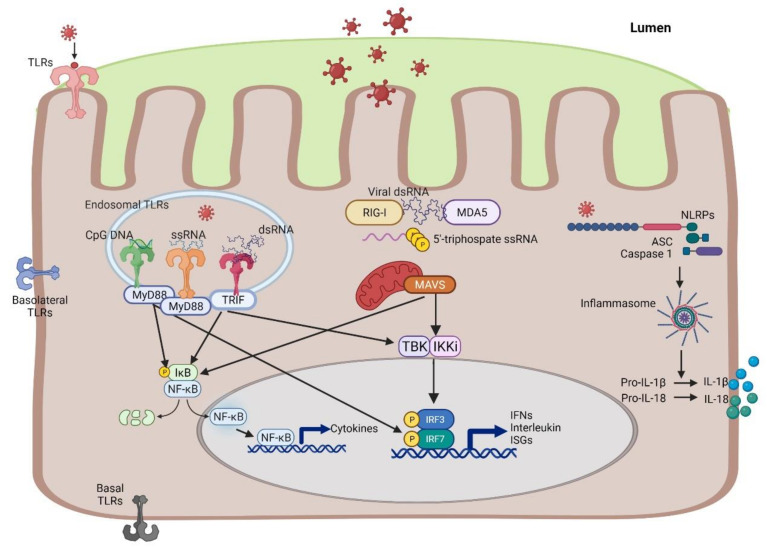
Schematic overview of PRRs involved in the sensing of viral infections in IECs. All TLRs share a common structure and signal through adaptor proteins. The elicited signaling pathway depends on the combination of TLRs and PAMPs. TLR2 and 4 recognize viruses at the cell surface, while TLR3 recognizes double-stranded (ds)RNA and TLR7 and 8 recognize single-stranded (ss)RNA, and TLR9 recognizes DNA with Unmethylated cytosine–guanine dinucleotide (CpG) motifs in the endosomes. Although it is not fully understood why, TLR3, TLR7, and TLR9 may occur both in the cell membrane and intracellularly [47]. TLRs may, in addition to apical and endosomal, include basolateral and basal cell membrane expression. All TLRs, except TLR3, utilize myeloid differentiation factor 88 (MyD88) as an adapter to recruit the signaling pathways leading to the activation of the transcription factors NF-kB, and in some cases interferon regulatory factor (IRF) 7, resulting in the production of pro-inflammatory cytokines and type I IFN. TLR3, on the other hand, utilizes the TIR domain-containing adaptor protein inducing interferon β (TRIF) to activate nuclear factor kappa-light-chain-enhancer of activated B cells (NF-kB) and IRF3 and can trigger the induction of pro-inflammatory cytokines and type I interferon β. The cytosolic receptors retinoic acid-inducible gene I (RIG-I) and melanoma differentiation-associated gene 5 (MDA5) detect viral-mediated dsRNA or ssRNA structures, respectively. They signal through the adaptor protein mitochondrial antiviral signaling (MAVS), eventually activating NF-kB or IRF3. NLRs are cytosolic PRR sensors that upon the recognition of PAMPs or DAMPs oligomerize into large macromolecular scaffolds called inflammasomes, which promote the maturation and secretion of interleukin (IL)-1β and IL-18. Illustration created by MDH using Biorender.com. https://biorender.com/ (accessed 30 August 2021).

## Data Availability

Not applicable.

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
