# Peer review of "Host–Viral Interactions in the Pathogenesis of Ulcerative Colitis"

_ijms, 2021, doi:10.3390/ijms221910851_

Round 1

Reviewer 1 Report

In this review, Ostvik and colleagues describes some recent studies about the interaction between viruses and the host on IBD pathogenesis. This is a very interesting and emerging topic now that lacks deep and detailed reviews. Although I appreciate that the authors put in a lot of effort in writing the manuscript and design the figures, I would propose to include some changes on the manuscript based on the following suggestions before it can be considered for publication.

Major points

The review is interesting, and it is well written in general, but the title account for a small part of the main text since the title claims about the host-viral interaction in ulcerative colitis while this is specifically described only in one paragraph, the number six.

In my opinion the authors should include more data about this interaction, or they should extend this part by an in-depth discussion of this relevant studies because this is the most novel and important advance of the review.

The review should include more discussion and increased perspective view from the authors, since they are experts in the topic.

Minor points

Please, add some text to Figure 2 legend

Please check English. In my opinion it is not well used the third tense of the verbs in some sentences:

For example “TLR3 on the other hand utilize” should be “TLR3s on the other hand utilize” or “TLR3 on the other hand utilizes”.

Other example, the text said “All TLRs share a common structure…” while some lines below it is said “All TLRs except TLR3 utilizes the myeloid differentiated…”

The punctuation could also be improved, in my opinion is easier to read “All TLRs, except TLR3, utilizes the myeloid differentiated…” than “All TLRs except TLR3 utilizes the myeloid differentiated…”

There are some typo mistakes as doble spaces around the manuscript and symbols in the IFN of the sentence “In line with studies from others[100,101], we showed that ISG15 enhanced release of IBD-relevant cy-tokines such as CXCL1, CXCL5, CXCL8, CCl20, IL1, IL6, TNF, IFN

Reviewer 2 Report

This is a very deep and detailed review of the role of viruses in the pathogenesis and the whole clinical picture of ulcerative colitis. It would be interesting to see as a short paragraph the clinical implications of this knowledge when it comes to diagnosing, preventing, and maybe possibly treating these viral infections. Thank you!
